# The Effect of Vaccination Rates on the Infection of COVID-19 under the Vaccination Rate below the Herd Immunity Threshold

**DOI:** 10.3390/ijerph18147491

**Published:** 2021-07-14

**Authors:** Yi-Tui Chen

**Affiliations:** Department of Health Care Management, National Taipei University of Nursing and Health Sciences, 89, Nei-Chiang St. Wan-Hua Dist., Taipei 108, Taiwan; yitui@ntunhs.edu.tw; Tel.: +886-2-23885111

**Keywords:** COVID-19, pandemic, herd immunity, turning point, vaccination rate, infection rate

## Abstract

Although vaccination is carried out worldwide, the vaccination rate varies greatly. As of 24 May 2021, in some countries, the proportion of the population fully vaccinated against COVID-19 has exceeded 50%, but in many countries, this proportion is still very low, less than 1%. This article aims to explore the impact of vaccination on the spread of the COVID-19 pandemic. As the herd immunity of almost all countries in the world has not been reached, several countries were selected as sample cases by employing the following criteria: more than 60 vaccine doses per 100 people and a population of more than one million people. In the end, a total of eight countries/regions were selected, including Israel, the UAE, Chile, the United Kingdom, the United States, Hungary, and Qatar. The results find that vaccination has a major impact on reducing infection rates in all countries. However, the infection rate after vaccination showed two trends. One is an inverted U-shaped trend, and the other is an L-shaped trend. For those countries with an inverted U-shaped trend, the infection rate begins to decline when the vaccination rate reaches 1.46–50.91 doses per 100 people.

## 1. Introduction

The pandemic of COVID-19 has hit the world for more than one year, and the number of new confirmed cases in many countries has reached a new high point in April–May 2021. As of 16 May 2021, the cumulative number of cases has reached 162,184,263, of which 4,809,520 new cases occurred from 10 to 16 May 2021 [1]. In the week of 16 May 2021 (10–16 May 2021), India reported to have 2,387,663 new cases, ranking at the top, ahead of Brazil (437,076 new cases) and the USA (235,638 new cases).

As of April 2021, several vaccines were authorized by at least one national regulatory authority for public use, including Pfizer–BioNTech, Moderna, BBIBP-CorV, CoronaVac, Covaxin, Sputnik V, AstraZeneca, and Johnson & Johnson [2]. Currently, vaccines produced by AstraZeneca, Pfizer–BioNTech, Sputnik V, Moderna, Sinopharm, Sinovac, and Johnson & Johnson have been widely used worldwide. As of 2 May 2021, 1.17 billion doses of COVID-19 vaccines have been administered in the world [3]. Vaccines may enhance the immune system to fight against the viruses and help an individual to prevent the infection of the pandemic, and, thus, many governments in the world are trying their full efforts to obtain sufficient vaccine supplies and speed up vaccination for the public.

However, the vaccine supply is limited due to insufficient capacity of vaccine production compared to the global demand. It is estimated that about 11 billion doses can vaccinate 70% of the world’s population to achieve herd immunity globally [4]. By the beginning of March 2021, about 413 million doses of COVID-19 vaccines have been produced, and this may increase to 9.5–12 billion doses by the end of 2021 [4]. The shortage of vaccine supply currently leads to uneven distribution of vaccine supply across countries. High- and upper-middle-income countries have secured more than 70% of the world’s COVID-19 vaccine doses [4]. The World Health Organization Chief emphasizes that 82% of vaccine doses were administered in high- or upper-middle-income countries, while only 0.2% had been sent to their low-income counterparts [5]. 

Vaccination can provide direct protection for the vaccinated individual to resist the attack of the virus and is a healthy way to limit the transmission of the pandemic through gaining population immunity [6,7]. However, it seems unlikely to achieve herd immunity in a short time, due to a currently limited supply of vaccines. Nevertheless, the reduction of epidemic transmission in a society may be essential to protect the unvaccinated individuals when a certain share of the population is vaccinated [8,9].

This paper attempts to examine the effect of vaccination on the infection rate of COVID-19. After reviewing data on new confirmed cases in association with vaccination rates in some countries with higher vaccination rates, this paper suggests that the infection rate after vaccination may have an inverted U-shaped trend. The infection rate rises and reaches its peak as the vaccination rate increases. After the peak (turning point), the infection rate starts to decline as vaccination rates increase.

Secondly, this paper also attempts to estimate how much vaccination rate is required to achieve the turning point. The solution of the turning point may help low-income or middle-income countries to determine the minimum vaccination rate, so as to avoid the spread of the COVID-19 pandemic, and design appropriate vaccination campaigns in the current global shortage of vaccines.

## 2. Materials and Methods

In this paper, the rolling weekly sum of new confirmed cases is used to calculate the weekly infection rate, $f_{t}$, that is defined as the ratio of new confirmed cases to population, expressed as: (1)$$f_{t}=\frac{n_{t}}{p}$$
where *n_t_* denotes the number of new confirmed cases in week *t*. 

To test the inverted U-shaped relationship between the vaccination rate and the infection rate of the COVID-19 epidemic, the square term of vaccination rates is incorporated into the model and serves as explanatory variables in addition to vaccination rates. Typically, individuals need two weeks after a one-dose vaccine or after vaccination of the second dose of a two-dose vaccine to start full protection against the COVID-19 virus [10]. The first dose of the vaccine may start to build up the process of the immune system, and the second dose aims to reinforce this protection [11]. The CDC suggests receiving the second dose within 6 weeks after the first dose [11]. Even though two doses are required for several vaccines, an individual still can get some immunity after receiving the first dose [11]. Considering both one-dose vaccines and two-dose vaccines are administered currently worldwide, this study employs two weeks to start producing antibody immunity. Thus, the infection rate $f_{t+2}$ is used as the dependent variable and is affected by the vaccination rate, $v_{t}$, that is calculated 2 weeks earlier. To test whether a turning point exists, the square, $v_{t}$, is also incorporated into the model, expressed as: (2)$$f_{t+2}=\alpha_{0}+\alpha_{1}v_{t}+\alpha_{2}v_{t}^{2}+e_{t}\;t=0,\;1,\;2,\ldots ,t,$$
where $v_{t}$ represents the vaccination rate,$\;e_{t}$ is the error term, $\beta_{0}$ is the intercept, and $\beta_{1}$ and $\beta_{2}$ are regression coefficients. Equation (2) describes that the new confirmed cases,$\;f_{t+2}$, as the dependent variable, is a function of the accumulated vaccination rate ($v_{t})$ and the square term of $v_{t}$. The analysis period starts one week before the vaccination starting date, denoted by *t* = 0, and the vaccination rate is 0. The analysis period ends on the week of 2 May 2021 (the last data for vaccination rates). Based on Equation (2), the corresponding week for the last data of infection rates is two weeks later than the data of vaccination rates and, thus, the last data of infection rates ends in the week of 16 May 2021. 

If an inverted U-turn exists, the coefficient $\beta_{2}$ should be negative. The peak (turning point) is obtained by taking a derivative of $f_{t}$ with respect to $v_{t-2}$ in Equation (2), yielding
(3)$$\frac{df_{t+2}}{dv_{t}}=\alpha_{1}+2\alpha_{2}{\bar{v}}_{t}=0\;$$
where ${\bar{v}}_{t}$ is the value of vaccination rates that reach the maximum of infection rates. Solving Equations (2) and (4) yields
(4)$${\bar{v}}_{t}=\frac{-\alpha_{1}}{2\alpha_{2}},\;$$
(5)$${\bar{f}}_{t+2}=\alpha_{0}-\frac{\alpha_{1}^{2}}{4\alpha_{2}}$$

And thus, the point of the peak (${\bar{v}}_{t-2}=\frac{-\alpha_{1}}{2\alpha_{2}}$, ${\bar{f}}_{t}$ = $\alpha_{0}-\frac{\alpha_{1}^{2}}{4\alpha_{2}}$) is obtained. 

### Data Collection 

The data for the number of new confirmed cases and total confirmed cases is extracted from the situation reports of World Health Organization [1], and the cumulative vaccination rate is provided from Our World in Data (2021). As this paper attempts to examine the trend of vaccination affecting infection rates and to determine the turning point of vaccination rates when herd immunity has not yet been achieved, the value of the accumulated vaccination rate should be high enough to cover the turning point. Therefore, it is recommended to use an accumulated vaccination rate of more than 50 doses per 100 people on the day of the last observation point as the criteria for selecting the sample country. 

As of 2 May 2021, a total of 20 countries have had a vaccination rate of more than 50 doses per 100 people. Gibraltar has the highest vaccination rate, which is 213.39 doses per 100 people, ahead of Seychelles (129.88 doses per 100 people). In order to have a more comprehensive understanding on the association of vaccination with infection rates, only countries with more than one million people are eligible to be selected as samples. Based on these two criteria, Israel, the United Arab Emirates (UAE), Chile, Bahrain, the United Kingdom (UK), the United States (USA), Hungary, Qatar, Uruguay, and Serbia were selected. However, the data on the vaccination rate of Bahrain during 13 January–5 March 2021 is not available. The vaccination for Uruguay started in 22 February 2021. The number of observation points for these two countries is not sufficient and, thus, Bahrain and Uruguay were excluded from the list of sample countries. Eventually, this study selected eight countries, including Israel, UAE, Chile, UK, USA, Hungary, Qatar, and Serbia, for analysis. 

The average weekly new confirmed cases (WNCC) during the analysis period, the accumulated vaccination rate (AVR) on 2 May 2021, and the vaccination starting date (VSD) for these sample countries is listed in Table 1. Due to different starting dates of vaccination among these countries, the number of observation points (NOP) is different for these countries and is indicated in Table 1. The weekly beginning and ending dates are determined based on the database of Coronavirus Disease (COVID-19) Weekly Epidemiological Update and Weekly Operational Update [1]. The data of new confirmed cases in each week is shown as the rolling weekly sum and is provided by WHO (2021). 

Table 1 demonstrates that an average of weekly new confirmed cases in the USA is the highest, reaching 794,700 cases each week, while Qatar has the lowest confirmed cases of 3400 during the analysis period. Israel’s accumulated vaccination rate is 120.85 doses per 100 people on 2 May 2021, ranking highest among these sample countries. Israel started its first dose of vaccinations on 20 December 2021, earlier than most countries. It took Israel a very short time to reach more than 50 doses per 100 people of the accumulated vaccination rate on 27 April 2021, and about 4 months to achieve 60% coverage rate (approximately 120 doses per 100 people) on 24 April 2021, while other countries needed more time to reach the same performance of vaccination. In contrast, Serbia spent about 4 months to reach the lowest vaccination rate of 51.83 doses only per 100 people on 2 May 2021. For example, in Qatar, the cumulative vaccination rate reached 44.13 doses per 100 people within 5 months (about 37% of Israel), while in the United States it reached 63.26 doses (about 53% of Israel). Among the eight countries, Israel, Chile, the USA, Hungary, and Qatar started to administer the vaccination before the end of 2020, while the rest started in the beginning of 2021. Each week is treated as an observation point, and thus the number of observation points ranges from 18 to 21. 

## 3. Results

The average weekly infection rate during the analysis period is listed in Table 2. Among these countries, the average weekly infection rate in Hungary ranked the highest, amounting to 24.84 cases per 10,000 people. In contrast, Qatar ranked at the bottom, having the lowest average weekly infection rate of 11.75 cases per 10,000 people. Among these countries, Israel had the highest value of the standard error of infection rates, while the UAE had the lowest. This implies that the variation of weekly infection rates across analysis periods was more in Israel than other countries. 

Table 2 demonstrates the weekly infection rate of Week 0 and the week of the last observation point. The reduction of infection rates from the peak and from Week 0 is also calculated and listed in Table 2. Compared to the peak, the weekly infection rate in the last observation point had been greatly reduced for all countries. Table 2 indicates that Israel performs the best improvement of infection rate from the peak, amounting to 67.03 cases per 10,000 people, while Chile only reduces its infection rates of 5.91 cases per 10,000 people from the peak. The weekly infection rate in Israel was reduced from the peak of 67.3 cases per 10,000 in the week of 17 January 2021 to 0.27 cases per 10,000 people in the week of 6 May 2021 (the last observation point). Chile decreased its weekly infection rate from the peak of 25.93 cases per 10,000 people in the week of 4 April 2021 to 20.02 cases per 10,000 people in the week of 16 May 2021 (the last observation point). 

Table 2 also indicates that the weekly infection rate in the last observation point has been greatly reduced from Week 0, except for Chile and Qatar. The UK gains the best performance, reducing its infection rates from 61.50 cases per 10,000 people in Week 0 to 2.32 cases per 10,000 people in the week of 16 May 2021 (the last observation point). The difference in infection rates between week 0 and the last observation point is negative in both Chile and Qatar, implying that infections in these two countries have increased in the last observation point compared to Week 0. In the week of the last observation point, the infection rate in Qatar and Chile was 8.07 and 20.02 cases per 10,000 people, higher than 3.63 cases and 5.91 per 10,000 people in Week 0.

### 3.1. The Trend of Infection Rates

The scatter diagram of $v_{t}$ versus $f_{t+2}$ during the analysis period (t=0, 1, 2,…,t, )  for each country is depicted in Figure 1. The comparison among countries in Figure 1 shows that, except for the UK, the infection rate in all countries increased from week 0 to a peak, and then declined. The infection rate in Week 0 was the peak in the UK, and then it kept a continual decreasing trend after vaccination until the week of the last observation point. The trend pattern of weekly infection rates in the USA was very similar to that in the UK. In the USA, the weekly infection rate increased from 40.31 cases per 10,000 people in Week 0 and 40.04 cases per 10,000 people in Week 1 to the peak of 53.98 cases per 10,000 people in Week 3. After Week 3, the weekly infection rate maintained a downward trend until the last observation point. 

### 3.2. The Effect of Vaccination on Infection Rates

The regression result of Equation (3) covering the analysis period in each country is listed in Table 3. The results show that the trends of infection rates after vaccination in Israel, UAE, Chile, Hungary, Qatar, and Serbia execute similar to an inverted U-shape, as the coefficient $\alpha_{2}\;$ for the term of square vaccination rates is significantly negative. The value of $\alpha_{2}\;$ indicated in Table 3 are −0.0053, −0.0011, −0.0046, −0.0403, −0.0165, and −0.0400 for these six countries, respectively. This phenomenon of the inverted U-shaped relationship between vaccination rates and infection rates implies that a peak exists after the start of vaccination and before the attainment of herd immunity. 

The theoretical peak of the fitted regression function for these six countries is calculated based on Equations (4) and (5). The turning point (peak) of the curve falls at the point of (27.14 doses per 100 people, 39.41 cases per 10,000 people) for Israel, (1.46 doses per 100 people, 22.37 cases per 10,000 people) for UAE, (50.91 doses per 100 people, 12.07 cases per 10,000 people) for Chile, (27.50 doses per 100 people, 14.46 cases per 10,000 people) for Hungary, (30.40 doses per 100 people, 5.32 cases per 10,000 people) for Qatar, and (24.90 doses per 100 people, 9.09 cases per 10,000 people) for Serbia. The accumulated vaccination rates of the theoretical peak happen at the range of 1.46–50.91 doses per 100 people. The R squared value of the estimation indicated in Table 3 ranges from 0.5718 to 0.8991, implying that the explanatory power is high, and the model of Equation (2) is appropriate to fit the data. 

However, the positive value of estimated $\;\alpha_{2}$ for the UK and the USA shows that the inverted U-shaped relationship between vaccination rates and infection rates does not exist. After reviewing the scatter diagrams of ($v_{t}$, $f_{t+2}$) indicated in Figure 1 for the UK and the USA, this paper proposes that an L-shaped relationship between vaccination and infection rates exists for these two countries. This paper conducted the Pearson correlation test and found that the coefficient is −0.7916 and −0.7315 for the UK and the USA, respectively. This means that higher vaccination rates in the UK and the USA led to lower infection rates during the whole analysis period, and the theoretical peak takes places when vaccination programs start. 

## 4. Discussion

This study attempts to investigate the effect of vaccination on the COVID-19 infection rates when the vaccination rate is below the herd immunity threshold. The estimation results shown in Table 3 show that the trend of infection rates after the start of vaccination follow two trends. One is an inverted U-shaped trend found in Israel, UAE, Chile, Hungary, Qatar, and Serbia, and the other is an L-shaped trend found in the UK and the USA. 

The pattern of an inverted U-shaped trend is characterized with an increasing infection rate after vaccination to a peak, and then turning to decline as the vaccination rate increases. The existence of the inverse U-turn of new confirmed cases may be explained by the lower vaccine effect below the turning point on reducing the infection. Only when the accumulated vaccination rate reaches a certain level does partial protection of herd immunity more or less take place, and the infection of disease is reduced. The study of Shen et al. [12] shows that low effective vaccines under low vaccination rates cannot suppress the spread of the COVID-19 pandemic. This paper concludes a similar viewpoint. The study of Shen et al. [12] developed a mathematical model to project the spread of the COVID-19 pandemic. It suggests that without using a mask, a 50% effective vaccine will not suppress infection at low vaccination coverage rate, but an 80% effective vaccine requires a 48–78% vaccination coverage rate, and a 100% effective vaccine requires 33–58% vaccination coverage to curb the spread of the COVID-19 pandemic. In the case of a mask usage rate of 50%, a 50% effective vaccine requires a 55–94% vaccination coverage rate, while an 80% effective vaccine only requires a 32–57% vaccination coverage rate, and a 100% effective vaccine requires a 24–46% vaccination coverage rate to suppress the spread of the pandemic. Thus, the gap of the reduction in infection rates across countries indicated in Table 2 may be attributable to the various vaccine effectiveness and vaccination rates. 

The regression results show that if the vaccination rate does not reach the vaccination threshold of 1.46–50.91 doses per 100 people, vaccination can only offset part of the spread of COVID-19, and the epidemic may worsen and resurge. This result is consistent with Zhang et al. [13], emphasizing that without the adequate protection of effective vaccines and the release of nonpharmaceutical interventions (NPIs), the epidemic may continue to rise; thus, this paper suggests that the cumulative vaccination rate of 1.46–50.91 doses per 100 people be used as the vaccination threshold of minimum requirements (VTMR) to avoid an exacerbation of the pandemic. Under the suggested VTMR, restrictive measures of NPIs should be mandatory to prevent further infection of the COVID-19 pandemic. 

The other pattern is found in the UK and the USA, executing similar to an L-shaped trend. After the first dose of vaccines administered in these two countries, the infection rate began to drop immediately. The cause of the L-shaped pattern may be explained by the fact that the infection rate just reached its peak before or when the vaccination program started.

Figure 2 depicts the number of weekly new confirmed cases in the UK and the USA from the four weeks before vaccination to the final observation point. In Figure 2, the numbers on the *x*-axis represent consecutive weeks after vaccination, where week 1 (*t =* 1) represents the week of the vaccination starting date, and the negative sign represents the week before the start of vaccination.

In the UK, the number of accumulated confirmed cases exceeded one million in the end of October 2020 and reached 1.6 million cases in the end of November 2020. The first data of vaccination rates in the UK started from 3 January 2021, according to the databank of Our World in Data (2021). The weekly new confirmed cases in the UK had kept a continual increasing trend since 4 weeks before the start of vaccination and reached a peak of 417,620 cases in week (−1), which represents one week before the start of vaccination. In the USA, Figure 2 indicates that the number of weekly new confirmed cases fluctuated before week 2 when a peak of 1,786,773 cases took place. After the peak, the number of new confirmed cases dropped sharply as the number of people vaccinated increased. 

In addition, both the United Kingdom and the United States have adopted strict measures of NPIs just before the start of vaccination to reduce human contact and to avoid mutual infection. From the beginning of December 2021, the UK implemented a three-tier system of tougher coronavirus restrictions to curb the spread of the pandemic [14]. On 4 January 2021 (week-1), the UK announced a third national lockdown for England with a legal requirement to stay at home. In contrast, the CDC issued a mandatory order requiring all passengers and operators of transportation hubs to wear masks to prevent the spread of infection on 9 January 2021, [15]. In week 2 (4–10 January 2021), the weekly new confirmed cases in the USA reached a peak. 

As the herd immunity threshold is proposed to be 65–70% of vaccine coverage rates, equivalent to about 130–140 doses per 100 people [16], it is not possible to return to normal life when less than 70% of the population in a country is immune, to keep the infection rate down. Generally, the herd immunity can be obtained either through vaccination administered or past exposure to the virus. Thus, the vaccination rate required to reach the herd immunity threshold may be less if confirmed cases are underreported. However, some scientists hold a pessimistic perspective to achieve a herd immunity threshold as the level of herd immunity depends on several factors, including the emergence of new variants, the delayed vaccination for children, and interpersonal contacts [17,18]. 

The virus may evolve to be more transmissible. Numerous new variants were detected in several countries in the past few months. One of these new variants, called B.1.1.7, was detected in the UK in September 2020 and spread to at least 114 countries in a few months. The study of Davis et al. [19] finds that the variant B.1.1.7 is 43% to 90% more transmissible than the original strain. In India, the new variant B.1.617, with two mutations, had led to a soaring increase in new confirmed cases and new deaths during April–May 2021. By April 2021, it had spread to at least 20 countries [20]. The study released by the Tata Institute of Fundamental Research and Indian Institute of Science estimates that the new variant detected in India is likely to be almost 200% to 250% more transmissible than the previous variant. The rapidly growing infection rate of COVID-19 in several countries during April–May 2021 may be attributed to the emergence of new variants [20]. Furthermore, it is also possible for a vaccinated individual to be infected again. At the end of April, a renowned scientist died of COVID-19, even though he had been vaccinated with two doses of the Pfizer vaccine in the USA [21]. 

The resurgence of the pandemic in the past few months implies that the effectiveness of vaccines to fight against the new variants detected in several countries, such as the Alpha variant (lineage B.1.1.7), the Beta variant (lineage B.1.351), the Gamma variant (lineage P.1), the Delta variant (lineage B.1.617), etc., are essential to reach herd immunity. Corey et al. [22] suggest that the epidemic can be prevented, and normal life can be returned to only if a highly effective vaccine is utilized for achieving herd immunity. The vaccine brands used in these countries are shown in Table 4. Among them, Pfizer/Biotech is more popular and has been widely used in all eight countries/regions. Moderna is the second-most widely accepted vaccine. As of 2 May 2021, approximately 130 million doses and 107 million doses of Pfizer/Biotech and Moderna vaccines have been vaccinated globally (Our World in Data, 2021). However, the amount of people vaccinated or the number of countries having used the vaccine to assess the effectiveness of the vaccine may cause errors. In the future, an analysis on the effectiveness of the used vaccines through the analysis of the real data may be focused.

If the effectiveness of vaccines against new variants decreases, the threshold value of vaccination coverage required to reach the herd immunity would increase. Gumel et al. [24] show that 82% of the vaccination coverage is required to support the realization of herd immunity through vaccination with 80% effective vaccines. If cumulative vaccination rates have not yet reached the herd immunity threshold, the premature release of prevention measures may trigger a relapse of the epidemic. If NPIs are released without sufficient protection from effective vaccines, the epidemic may worsen and resurge [13]. To curb the spread of the pandemic and the reopening of the economy, the implementation of the vaccination program as early as possible is essential [25]. Furthermore, this paper suggests that the implementation of the vaccination program is only one factor leading to the successful decline of the infection rate. Another factor of non-pharmaceutical interventions, such as wearing masks, maintaining social distancing, large-scale virus testing, and public cooperation, as well as vaccine effectiveness and vaccination coverage, have also played an important role in preventing the spread of the pandemic.

## 5. Conclusions

This study found that, based on data from these countries with higher vaccination rates, the two infection rate trends after the start of vaccination include an inverted U-shape and an L-shape. This result implies that all vaccines currently administered in these countries are highly effective in mitigating the infection of COVID-19 when vaccination rates reach the level of turning points, even if the vaccination rate is lower than the herd immunity threshold. This paper suggests that the value of vaccination rate of 50.91 doses per 100 people is a minimum requirement for avoiding the aggravation of the pandemic in a society, and, thus, the vaccination program should follow an intensive manner to assure the attainment of the turning point in a short time to prevent the resurgence of COVID-19 infection. The results of this study can be integrated into existing policies in response to the attack of the COVID-19 pandemic. 

To support the herd immunity threshold, administering the vaccination on a sufficient proportion of the population is required. Only above the herd immunity threshold can the transmission can be fully stopped. It is possible to reach local herd immunity in some continents, countries, or states, but it is very difficult to achieve long-term, global herd immunity when considering the low coverage rate of vaccination in Africa and many poor countries. It may remain a challenge for policymakers to increase the coverage rate of vaccination as quickly as possible in the world. Vaccination plays a key role in fighting against the pandemic of COVID-19, but it depends on the power of vaccines to resist the attack of new variants. This paper suggests that vaccines are not a panacea in solving the pandemic of COVID-19 if the new variant provides a significant impact on transmissibility, severity, and/or immunity. If individuals currently vaccinated are proven to lack resistance to new variants, new vaccines should be developed and, thus, the pandemic is unlikely to be stopped in a short time.

## Figures and Tables

**Figure 1 ijerph-18-07491-f001:**
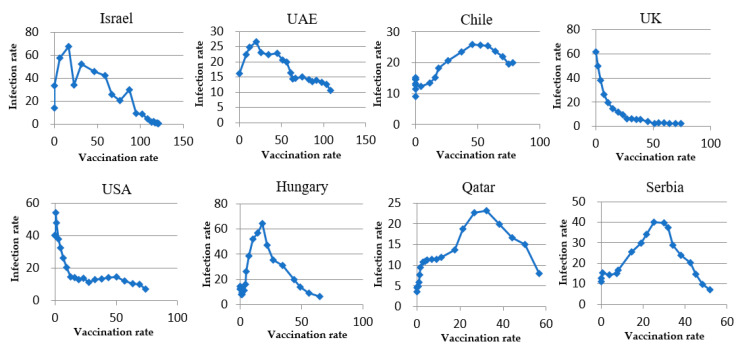
The scatter diagram of vaccination rates vs. infection rates (2 weeks later) for each country.

**Figure 2 ijerph-18-07491-f002:**
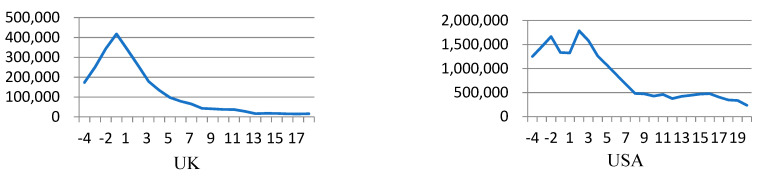
The trend of weekly new confirmed cases in the UK and the USA.

**Table 1 ijerph-18-07491-t001:** The descriptive statistics among countries selected.

Country	Israel	UAE	Chile	UK	USA	Hungary	Qatar	Serbia
Average WNCC	19,500	18,000	32,900	114,800	794,700	24,300	3400	
AVR	120.85	107.53	77.87	73.80	73.43	63.67	55.67	51.83
VSD	20/12/20	21/01/05	20/12/25	21/01/03	20/12/20	20/12/28	20/12/22	21/01/08
NOP	21	18	20	19	21	20	20	18

**Table 2 ijerph-18-07491-t002:** Infection rates during the analysis period (unit: cases per 10,000 people).

	Israel	UAE	Chile	UK	USA	Hungary	Qatar	Serbia
Average	21.56	17.73	17.83	14.37	21.95	24.84	11.75	22.01
Standard error	463.89	23.10	28.17	307.45	199.23	334.17	34.30	114.41
Max. (peak)	67.30	26.58	25.93	61.52	53.98	64.45	23.23	39.88
Min.	0.27	10.60	9.16	2.14	7.12	6.66	3.63	7.23
Week 0	14.26	16.24	9.16	61.52	40.31	12.35	3.63	10.98
Week (last)	0.27	10.60	20.02	2.32	7.12	6.66	8.07	7.23
Diff. of Week 0—peak	67.03	15.98	5.91	59.20	46.86	57.79	15.16	32.65
Diff. of Week 0—last	13.99	5.64	−10.86	59.20	33.19	5.69	−4.44	3.75

**Table 3 ijerph-18-07491-t003:** The regression result of Equation (2).

	Israel	UAE	Chile	UK	USA	Hungary	Qatar	Serbia
$\alpha_{0}$	39.41 *** (5.91)	22.37 *** (1.71)	12.07 *** (0.73)	47.05 *** (3.42)	40.19 *** (2.75)	14.41 *** (4.48)	5.32 *** (0.67)	9.05 *** (2.15)
$\alpha_{1}$	0.2891 (0.2377)	0.0032 (0.0705)	0.4688 *** (0.0638)	−1.92 *** (0.2335)	−1.36 *** (0.2254)	2.22 *** (0.4971)	1.00 *** (0.0863)	1.99 *** (0.2178)
$\alpha_{2}$	−0.0053 *** (0.0018)	−0.0011 * (0.0006)	−0.0046 *** (0.0009)	0.0189 *** (0.0032)	0.0138 *** (0.0032)	−0.0403 *** (0.0085)	−0.0165 *** (0.0017)	−0.0400 *** (0.0043)
R2	0.7698	0.7275	0.8629	0.8813	0.7859	0.5718	0.8991	0.8512
ob.	21	18	20	19	21	20	20	18

(): standard errors, *: *p* ≤ 0.10, ***: *p* ≤ 0.01, ob.: the number of observation points.

**Table 4 ijerph-18-07491-t004:** The brand/manufacturers of vaccines used in these countries.

Country	Vaccine Brands/Manufacturers
Israel ^#1^	Pfizer/Biotech; Moderna
UAE ^#1^	Pfizer/Biotech; Sinopharm; Sputnik V; Oxford/AstraZeneca
Chile ^#2^	Pfizer/Biotech; Oxford/AstraZeneca; Sinovac
UK ^#1^	Pfizer/Biotech; Moderna; Johnson & Johnson; Oxford/AstraZeneca
USA ^#2^	Pfizer/Biotech; Moderna; Johnson & Johnson
Hungary ^#2^	Pfizer/Biotech; Moderna; Oxford/AstraZeneca; Sputnik V; Johnson & Johnson
Serbia ^#1^	Pfizer/Biotech; Sinopharm; Sputnik V; Oxford/AstraZeneca
Qatar ^#1^	Pfizer/Biotech; Moderna

^#1^ Source: COVID-19 Vaccine Tracker [23]. ^#2^ Source: Our World in Data [3].

## Data Availability

The data for COVID-19 confirmed cases and vaccination rates can be found in WHO (https://www.who.int/emergencies/diseases/novel-coronavirus-2019/situationreports (accessed on 24 May 2021)) and Our World in Data (https://ourworldindata.org/covidvaccinations (accessed on 24 May 2021)), respectively.

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
