# Peer review of "The Effect of Vaccination Rates on the Infection of COVID-19 under the Vaccination Rate below the Herd Immunity Threshold"

_ijerph, 2021, doi:10.3390/ijerph18147491_

Round 1
Reviewer 1 Report
Although the subject and aims are interesting and important, the paper shows serious flaws.
It aims to demonstrate the impact of vaccination, but results focuses more on associations and not enough was done to demonstrate cause-effect.
The presentation of results is not adequate, mainly in what regards graphics.
A more thorough and sound bibliographical research should have been made to support the discussion and, thus, conclusions.
Author Response
Responses to Reviewers’ comments
Response to Reviewer 1:
Comment: It aims to demonstrate the impact of vaccination, but results focuses more on associations and not enough was done to demonstrate cause-effect.
Response: Thank you very much for your valuable comments. In fact, the cumulative vaccination rates of all countries selected in this article were only analyzed until May 2, 2021, which was still very low and far below the herd immunity threshold. In the case of low vaccination rates or low vaccine effectiveness, vaccination can only partially offset the spread of the pandemic (Shen et al., 2011). And thus, it is very difficult to analyze the causal effect of vaccination due to low level of vaccination rates. This paper attempts to examine only on the outcome of the reduced infection rate under the vaccination rate below the herd immunity threshold. In this case, the title of this paper was revised to:
"The effect of vaccination rates on the infection of COVID-19 under the vaccination rate below the herd immunity threshold”
Reference:
Shen, M.; Zu, J.; Fairley, C. K.; Pagán, J. A.; An, L.; Du, Z.; Guo, Y.; Rong, L.; Xiao, Y.; Zhuang, G.; Li, Y.; Zhang, L. 2021. Projected COVID-19 epidemic in the United States in the context of the effectiveness of a potential vaccine and implications for social distancing and face mask use. Vaccine 2021, 39(16), 2295-2302.
Comment: The presentation of results is not adequate, mainly in what regards graphics.
Response: Thank you very much for your comment. The text space for Figure 1-8 contains more than 2 pages, which seems inappropriate. In order to focus on the result in association with the research purpose including the analysis on the reduction in the infection rates, the examination on the inverted U-shaped trend of infection rates and the determination of the turning point, this paper added a sub-section after the paragraph starting “Table 2 also indicates that the weekly infection rate …higher than 3.63 cases and 5.91 per 10,000 people in Week 0.” in “3. Results” on Page. 4. Figure 1-8 were also combined together into a figure and the image was shrunk to reduce the text space, expressed as:
“Table 2 also indicates that the weekly infection rate … higher than 3.63 cases and 5.91 per 10,000 people in Week 0.”
3.1 The trend of infection rates
The scatter diagram of versus during the analysis period ( for each country is depicted in Figure 1.… till the last observation point.”
Comment: A more thorough and sound bibliographical research should have been made to support the discussion and, thus, conclusions.
Response: Thank you very much for your comment. Several new paragraphs were inserted to follow your valuable comments, including:
On Page 6 in the revised version of this article, the paragraph starting “The pattern of an inverted U-shaped trend is characterized with …” was revised, reading as:
“The pattern of an inverted U-shaped trend is characterized with …and the infection of disease is reduced. The study of Shen et al (2021) shows that low effective vaccines under low vaccination rates cannot suppress the spread of the COVID-19 pandemic. This paper concludes the similar view point. The study of Shen et al. (2021) develop a mathematical model to project the spread of the COVID-19 pandemic. It suggests that without using a mask, a 50% effective vaccine will not suppress infection at low vaccination coverage rate, but an 80% effective vaccine requires a 48-78% vaccination coverage rate, and a 100% effective vaccine requires 33-58% vaccination coverage to curb the spread of the COVID-19 pandemic. In the case of a mask usage rate of 50%, a 50% effective vaccine requires a 55-94% vaccination coverage rate, while an 80% effective vaccine only requires a 32-57% vaccination coverage rate and a 100% effective vaccine requires a 24-46% vaccination coverage rate to suppress the spread of the pandemic. Thus, the gap of the reduction in infection rates across countries indicated in Table 2 may be attributable to the various vaccine effectiveness and vaccination rates.
The regression result shows that if the vaccination rate does not reach the vaccination threshold of 1.46-50.91 doses per 100 people, vaccination can only offset part of the spread of COVID-19 and the epidemic may worsen and resurge. This result is consistent with Zhang et al. (2020), emphasizing that without the adequate protection of effective vaccines and the release of non-pharmaceutical interventions (NPIs), the epidemic may continue to rise. And thus, this paper suggests that the cumulative vaccination rate of 1.46-50.91 doses per 100 people be used as the vaccination threshold of minimum requirements (VTMR) to avoid an exacerbation of the pandemic. Under the suggested VTMR, restrictive measures of NPIs should be mandatory to prevent further infection of the COVID-19 pandemic.”
On Page 8 in the revised version of this article, the last paragraph in “4. Discussions” section was revised, reading as:
“If the effectiveness of vaccines against new variants decreases, the threshold value of vaccination coverage required to reach the herd immunity would increase. Gumel et al. (2020) show that 82% of the vaccination coverage is required to support the realization of herd immunity through vaccination with an 80% effective vaccines. If cumulative vaccination rates have not yet reached the herd immunity threshold, the premature release of prevention measures may trigger a relapse of the epidemic. If NPIs are released without sufficient protection from effective vaccines, the epidemic may worsen and resurge (Zhang et al., 2020). To curb the spread of the pandemic and the reopening of the economy, the implementation of the vaccination program as early as possible is essential (Adalja et al., 2020). Furthermore, this paper suggests that the implementation of the vaccination program is only one factor leading to the successful decline of the infection rate. Another factor of non-pharmaceutical interventions such as wearing masks, maintaining social distancing, large-scale virus testing and public cooperation, as well as vaccine effectiveness and vaccination coverage have also played an important role in preventing the spread of the pandemic.”
Reference:
Adalja, A.A.; Toner, E.; Inglesby, T.V. Priorities for the US health community responding to COVID-19. JAMA 2020, 323(14), 1343-1344.
Corey, L.; Mascola, J.R.; Fauci, A.S.; Collins, F.S. A strategic approach to COVID-19 vaccine R&D. Science 2020, 368(6494), 948-950.
Gumel, A.B.; Iboi, E.A.; Ngonghala, C.N.; Will an imperfect vaccine curtail the COVID19 pandemic in the US?. Infectious Disease Modelling 2020, 5, 510-524.
Shen, M.; Zu, J.; Fairley, C. K.; Pagán, J. A.; An, L.; Du, Z.; Guo, Y.; Rong, L.; Xiao, Y.; Zhuang, G.; Li, Y.; Zhang, L. 2021. Projected COVID-19 epidemic in the United States in the context of the effectiveness of a potential vaccine and implications for social distancing and face mask use. Vaccine 2021, 39(16), 2295-2302.
Zhang, L.; Tao, Y.; Shen, M.; Fairley, C.K.; Guo, Y. Can self-imposed prevention measures mitigate the COVID-19 epidemic?. PLoS Med 2020, 17(7), e1003240.

Reviewer 2 Report
Chen used available vaccine data and COVID-19 infection data to evaluate the relation of the efficacy of vaccine against COVID-19 infection. It is really interest and could be of interest to general readers. However, I would like to point out two major issues. First, the definition of two types of outcome after vaccine, the reverse"U" shape and the "L" shape, is not very accurate. If the vaccine works, it should be "L" shape. If it does not work effectively or even no vaccine, it should be reverse "U" shaped. Second, from analysis, it is obvious that two of eight, Chile and UAE, the drop of infection is very slow or does not drop at all, it suggests that the vaccine is not effective. Qutar drops slowly after vaccine. The vaccine is questionable. Third, you may need to specify vaccines used for each country.
Author Response
Responses to Reviewers’ comments
Response to Reviewer 2:
Comment: I would like to point out two major issues. First, the definition of two types of outcome after vaccine, the reverse "U" shape and the "L" shape, is not very accurate. If the vaccine works, it should be "L" shape. If it does not work effectively or even no vaccine, it should be reverse "U" shaped.
Response: Thank you very much for your comment. In “4. Discussions”, this paper has discussed the possible cause for the formation of the reverse “U” shape and the “L” shape. The result of this paper points out that the spread of COVID-19 may start to decrease only after the cumulative vaccination rate is beyond a certain level. The low vaccination coverage rate or low effective vaccines would not suppress the spread of the infection (Shen et al., 2021). This paper suggests the certain level to be 1.46-50.91 doses per 100 people and terms it as the vaccination threshold of minimum requirements (VTMR) to avoid an exacerbation of the pandemic. The peak of the "L"-shaped trend occurred just before and after the start of vaccination, as experienced in the United Kingdom and the United States. The variation of the infection rate may be caused by another factor like non-pharmaceutical interventions in case of the cumulative vaccination rate below VTMR. To have a more comprehensive understanding, two new paragraphs were inserted after the paragraph starting “The pattern of an inverted U-shaped trend is characterized …” on P. 6 in the revised version of this article, reading as:
“The pattern of an inverted U-shaped trend is characterized … and the infection of disease is reduced. The study of Shen et al. (2021) shows that low effective vaccines under low vaccination rates cannot suppress the spread of the COVID-19 pandemic. This paper concludes the similar view point. The study of Shen et al. (2021) develop a mathematical model to project the spread of the COVID-19 pandemic. It suggests that without using a mask, a 50% effective vaccine will not suppress infection at low vaccination coverage rate, but an 80% effective vaccine requires a 48-78% vaccination coverage rate, and a 100% effective vaccine requires 33-58% vaccination coverage to curb the spread of the COVID-19 pandemic. In the case of a mask usage rate of 50%, a 50% effective vaccine requires a 55-94% vaccination coverage rate, while an 80% effective vaccine only requires a 32-57% vaccination coverage rate and a 100% effective vaccine requires a 24-46% vaccination coverage rate to suppress the spread of the pandemic. Thus, the gap of the reduction in infection rates across countries indicated in Table 2 may be attributable to the various vaccine effectiveness and vaccination rates.
The regression result shows that if the vaccination rate does not reach the vaccination threshold of 1.46-50.91 doses per 100 people, vaccination can only offset part of the spread of COVID-19 and the epidemic may worsen and resurge. This result is consistent with Zhang et al. (2020), emphasizing that without the adequate protection of effective vaccines and the release of non-pharmaceutical interventions (NPIs), the epidemic may continue to rise. And thus, this paper suggests that the cumulative vaccination rate of 1.46-50.91 doses per 100 people be used as the vaccination threshold of minimum requirements (VTMR) to avoid an exacerbation of the pandemic. Under the suggested VTMR, restrictive measures of NPIs should be mandatory to prevent further infection of the COVID-19 pandemic.”
Comment: Second, from analysis, it is obvious that two of eight, Chile and UAE, the drop of infection is very slow or does not drop at all, it suggests that the vaccine is not effective. Qatar drops slowly after vaccine. The vaccine is questionable. Third, you may need to specify vaccines used for each country.
Response: Thank you very much. The vaccine may be questionable but no evidence to show it. We follow your suggestion and the adopted vaccine for use in these countries is mentioned in the revised version of this paper. Furthermore, your comment is valuable, and it is worth discussing the reasons for the decline in infection rates in these three countries. The paragraph starting “The pattern of an inverted U-shaped trend is characterized with an increasing … “ in the “4. Discussions” section on P. 6 was revised, reading as:
“The pattern of an inverted U-shaped trend is characterized … and the infection of disease is reduced. … The study of Shen et al. (2021) develop a mathematical model to project the spread of the COVID-19 pandemic. It suggests that without using a mask, a 50% effective vaccine will not suppress infection at low vaccination coverage rate, but an 80% effective vaccine requires a 48-78% vaccination coverage rate, and a 100% effective vaccine requires 33-58% vaccination coverage to curb the spread of the COVID-19 pandemic. In the case of a mask usage rate of 50%, a 50% effective vaccine requires a 55-94% vaccination coverage rate, while an 80% effective vaccine only requires a 32-57% vaccination coverage rate and a 100% effective vaccine requires a 24-46% vaccination coverage rate to suppress the spread of the pandemic. Thus, the gap of the reduction in infection rates across countries indicated in Table 2 may be attributable to the various vaccine effectiveness and vaccination rates.
On Page 8 in the revised version of this article, a new paragraph together with a new table was inserted after the paragraph starting “The virus may evolve to…” to discuss the vaccine used in these countries, reading as:
“The virus may evolve to… two doses of the Pfizer vaccine in the USA [19].” “The resurgence of the pandemic in the past few months implies that the effectiveness of vaccines to fight against the new variants detected in several countries such as the Alpha variant (lineage B.1.1.7), the Beta variant (lineage B.1.351), the Gamma variant (lineage P.1), the Delta variant (lineage B.1.617), etc. are essential to reach herd immunity. Corey et al. (2020) suggest that the epidemic can be prevented and the normal life can be returned only if a highly effective vaccine is vaccinated for achieving herd immunity. The vaccine brands used in these countries are shown in Table 4. Among them, Pfizer/Biotech is more popular and has been widely used in all 8 countries/regions. Moderna is the second widely accepted vaccine. As of May 2, 2021, approximately 130 million doses and 107 million doses of Pfizer/Biotech and Moderna vaccines have been vaccinated globally (Our World in Data, 2021). However, the amount of vaccine vaccinated or the number of countries having used the vaccine to assess the effectiveness of the vaccine may cause errors. In the future, an analysis on the effectiveness of the used vaccines through the analysis of the real data may be focused.
Table 4. The brand/manufacturers of vaccines used in these countries
|
country |
Vaccine brands/manufacturers |
|
Israel#1 |
Pfizer/Biotech; Modern |
|
UAE#1 |
Pfizer/Biotech; Sinopharm; Sputnik V; Oxford/AstraZeneca |
|
Chile#2 |
Pfizer/Biotech; Oxford/AstraZeneca; Sinovac |
|
UK#1 |
Pfizer/Biotech; Modern; Johnson & Johnson; Oxford/AstraZeneca |
|
USA#2 |
Pfizer/Biotech; Modern; Johnson & Johnson |
|
Hungary#2 |
Pfizer/Biotech; Modern; Oxford/AstraZeneca; Sputnik V; Johnson & Johnson |
|
Serbia#1 |
Pfizer/Biotech; Sinopharm; Sputnik V; Oxford/AstraZeneca |
|
Qatar#1 |
Pfizer/Biotech; Modern |
#1 Source: COVID-19 Vaccine Tracker [23]
#2 Source: Our World in Data [3]
Reference:
Corey, L.; Mascola, J.R.; Fauci, A.S.; Collins, F.S. A strategic approach to COVID-19 vaccine R&D. Science 2020, 368(6494), 948-950.
Our World in Data. Coronavirus (COVID-19) vaccinations. 2021. Available online: https://ourworldindata.org/covid-vaccinations (accessed on 21 June 2021).
Shen, M.; Zu, J.; Fairley, C. K.; Pagán, J. A.; An, L.; Du, Z.; Guo, Y.; Rong, L.; Xiao, Y.; Zhuang, G.; Li, Y.; Zhang, L. 2021. Projected COVID-19 epidemic in the United States in the context of the effectiveness of a potential vaccine and implications for social distancing and face mask use. Vaccine 2021, 39(16), 2295-2302.
